# An analytic theory of shallow networks dynamics for hinge loss classification

**Franco Pellegrini**
Laboratoire de Physique de l'École normale supérieure, ENS,
Université PSL, CNRS, Sorbonne Université, Université de Paris
F-75005 Paris, France
`franco.pellegrini@phys.ens.fr`

**Giulio Biroli**
Laboratoire de Physique de l'École normale supérieure, ENS,
Université PSL, CNRS, Sorbonne Université, Université de Paris
F-75005 Paris, France

## Abstract

Neural networks have been shown to perform incredibly well in classification tasks over structured high-dimensional datasets. However, the learning dynamics of such networks is still poorly understood. In this paper we study in detail the training dynamics of a simple type of neural network: a single hidden layer trained to perform a classification task. We show that in a suitable mean-field limit this case maps to a single-node learning problem with a time-dependent dataset determined self-consistently from the average nodes population. We specialize our theory to the prototypical case of a linearly separable data and a linear hinge loss, for which the dynamics can be explicitly solved in the infinite dataset limit. This allows us to address in a simple setting several phenomena appearing in modern networks such as slowing down of training dynamics, crossover between rich and lazy learning, and overfitting. Finally, we assess the limitations of mean-field theory by studying the case of large but finite number of nodes and of training samples.

## 1 Introduction

Despite their proven ability to tackle a large class of complex problems [1], neural networks are still poorly understood from a theoretical point of view. While general theorems prove them to be universal approximators [2], their ability to obtain generalizing solutions given a finite set of examples remains largely unexplained. This behavior has been observed in multiple settings. The huge number of parameters and the optimization algorithms employed to optimize them (gradient descent and its variations) are thought to play key roles in it [3–5].

In consequence, a large research effort has been devoted in recent years to understanding the training dynamics of neural networks with a very large number of nodes [6–8]. Much theoretical insight has been gained in the training dynamics of linear [9, 10] and nonlinear networks for regression problems, often with quadratic loss and in a teacher-student setting [11–14], highlighting the evolution of correlations between data and network outputs. More generally, the input-output correlation and its effect on the landscape has been used to show the effectiveness of gradient descent [15, 16]. Other approaches have focused on infinitely wide networks to perform a mean-field analysis of the weights dynamics [17–22], or study its neural tangent kernel (NTK, or "lazy") limit [23–26].

In this work, we investigate the learning dynamics for binary classification problems, by considering one of the most common cost functions employed in this setting: the linear hinge loss. The idea behind the hinge loss is that examples should contribute to the cost function if misclassified, but also if classified with a certainty lower than a given threshold. In our case this cost is linear in the distance from the threshold, and zero for examples classified above threshold, that we shall call *satisfied* henceforth. This specific choice leads to an interesting consequence: the instantaneous gradient for each node due to *unsatisfied* examples depends on the activation of the other nodes only through their population, while that due to *satisfied* examples is just zero. Describing the learning dynamics in the mean-field limit amounts to computing the effective example distribution for a given distribution of parameters: each node then evolves "independently" with a time-dependent dataset determined self-consistently from the average nodes population.

**Contribution.** We provide an analytical theory for the dynamics of a single hidden layer neural network trained for binary classification with linear hinge loss. In Sec. 2 we obtain the mean-field theory equations for the training dynamics. Those equations are a generalizations of the ones obtained for mean-square loss in [17–22]. In Sec. 3 we focus on linearly separable data with spherical symmetry and present an explicit analytical solution of the dynamics of the nodes parameters. In this setting we provide a detailed study of the cross-over between the lazy [23] and rich [27] learning regimes (Sec. 3.2). Finally, we assess the limitations of mean-field theory by studying the case of large but finite number of nodes and finite number of training samples (Sec. 3.3). The most important new effect is overfitting, which we are able to describe by analyzing corrections to mean-field theory. In Sec. 3.4 we show that introducing a small fraction of mislabeled examples induces a slowing down of the dynamics and hastens the onset of the overfitting phase. Finally in Sec. 4 we present numerical experiments on a realistic case, and show that the associated nodes dynamics in the first stage of training is in good agreement with our results.

The merit of the model we focused on is that, thanks to its simplicity, several effects happening in real networks can be studied analytically. Our analytical theory is derived using reasoning common in theoretical physics, which we expect can be made rigorous following the lines of [17–22]. All our results are tested throughout the paper by numerical simulations which confirm their validity.

**Related works.** The study of neural network dynamics with one (or few) nodes started in statistical physics [11], but was mainly focused on the online setting. More recent works on separable data [28, 29] observed the main trend of logarithmic alignment with the max margin vector under rather general settings. Mean-field analysis of the training dynamics of very wide neural networks have mainly focused on regression problems with mean-square losses [17–23], whereas fewer works [30, 31] have tackled the dynamics for classification tasks.[1] The task and architecture we focus on bears strong similarities to the one proposed in des Combes et al. [30], but with fewer assumptions on the dataset and initialization. With respect to [30], we show the relation with mean-field treatments [17–22] and provide a full analysis of the dynamics, in particular the cross-over between rich and lazy learning. Moreover, we discuss the limitations of mean-field theory, the source of overfitting and the change in the dynamics due to mislabeling.

## 2 Mean-Field equation for the density of parameters

We consider a binary classification task for $N$ points in $d$ dimensions $\{\mathbf{x}_n\} \subset \mathbb{R}^d$ with corresponding labels $y_n = \pm 1$. We focus on a hidden layer neural network consisting of $M$ nodes with activation $\sigma$. The output of the network is therefore

$$f(\mathbf{x}; \boldsymbol{\theta}) = \frac{1}{M} \sum_{i=1}^{M} a_i \sigma \left( \frac{\mathbf{w}_i \cdot \mathbf{x}}{\sqrt{d}} \right), \tag{1}$$

where $\boldsymbol{\theta}_i = \{a_i, \mathbf{w}_i\}$ represents all the trainable parameters of the model: $\{\mathbf{w}_i\}$, the $d$-dimensional weight vectors between input and each hidden node, and $\{a_i\}$, the contributions of each node to the output. All components are initialized before training from a Gaussian distribution with zero mean and unit standard deviation. The $1/M$ in front of the sum leads to the so-called mean-field normalization [17]. In the large-$M$ limit, this allows to do what is called a hydrodynamic treatment in physics, a procedure that have been put on a rigorous basis in this context in [17–23] (here the $\boldsymbol{\theta}_i$s play the role of particle positions). One of the main assumptions of this procedure is that in the large

$M$-limit one can rewrite the output function in terms of the averaged nodes population (or density) $\rho(\boldsymbol{\theta})$:

$$f(\mathbf{x}; \boldsymbol{\theta}) = \int d\boldsymbol{\theta} \rho(\boldsymbol{\theta}) a\sigma \left( \frac{\mathbf{w} \cdot \mathbf{x}}{\sqrt{d}} \right). \tag{2}$$

To optimize the parameters we minimize the loss function

$$\mathcal{L} = \frac{1}{N} \sum_{n=1}^{N} \ell(y_n, f(\mathbf{x}_n; \boldsymbol{\theta})) \tag{3}$$

by gradient flow $\dot{\boldsymbol{\theta}} = -\beta^* \partial \mathcal{L} / \partial \boldsymbol{\theta}$ ($\ell(x, y)$ will be specified later). The dynamical equations for the parameters $\{a_i, \mathbf{w}_i\}$ read:

$$\begin{cases} \dot{a}_i &= -\dfrac{\beta}{N} \sum_{n=1}^{N} \dfrac{\partial \ell(y_n, f(\mathbf{x}; \boldsymbol{\theta}))}{\partial f} \sigma \left( \dfrac{\mathbf{w}_i \cdot \mathbf{x}}{\sqrt{d}} \right) \\ \dot{\mathbf{w}}_i &= -\dfrac{\beta}{N} \sum_{n=1}^{N} \dfrac{\partial \ell(y_n, f(\mathbf{x}; \boldsymbol{\theta}))}{\partial f} a_i \sigma' \left( \dfrac{\mathbf{w}_i \cdot \mathbf{x}}{\sqrt{d}} \right) \dfrac{\mathbf{x}}{\sqrt{d}}, \end{cases} \tag{4}$$

where we have defined the effective learning rate $\beta = \beta^*/M$. These equations show that the coupling between the different nodes has a mean-field form: it is through the function $f$, i.e. only through the density $\rho(\boldsymbol{\theta}, t)$. Following standard techniques one can obtain a closed hydrodynamic-like equation on $\rho(\boldsymbol{\theta}, t)$ in the large $M$ limit:

$$\partial_t \rho(\boldsymbol{\theta}, t) = \beta \nabla_{\boldsymbol{\theta}} \left( \rho(\boldsymbol{\theta}, t) \nabla_{\boldsymbol{\theta}} \frac{\delta \mathcal{L}[\rho(\boldsymbol{\theta}, t)]}{\delta \rho(\boldsymbol{\theta}, t)} \right) \; , \; \rho(\boldsymbol{\theta}, 0) = \mathcal{N}(0, \mathbb{I}) \tag{5}$$

where we have made explicit that the $\mathcal{L}$ is a functional of the density $\rho$ since it depends on $f(\mathbf{x}; \boldsymbol{\theta})$, see eqs. (2, 3). The convergence of the dynamical process to the hydrodynamic limit is usually assumed in the physics literature, proofs (that we expect can be generalized to our case) have been worked out in [32, 33]. (see SM for details)

To be more concrete, in the following we consider the case of linear hinge loss, $\ell(y, f) = \mathcal{R}(h - yf)$ ($h$ being the size of the hinge, often taken as 1), and rectified linear unit (ReLU) activation function: $\sigma(x) = \mathcal{R}(x) = \max(0, x)$. With this choice

$$\frac{\delta \mathcal{L}[\rho(\boldsymbol{\theta}, t)]}{\delta \rho(\boldsymbol{\theta}, t)} = -a \left\langle u(\mathbf{x}, y; t) \theta (\mathbf{w} \cdot \mathbf{x}) y \frac{\mathbf{w} \cdot \mathbf{x}}{\sqrt{d}} \right\rangle_{\mathbf{x}, y}, \tag{6}$$

$\theta$ being the Heaviside step function. The notation $u(\mathbf{x}, y; t) \equiv \mathbb{I}_{h - yf(\mathbf{x}; \boldsymbol{\theta}(t)) > 0}$ denotes the indicator function of the *unsatisfied* examples, i.e. those $(\mathbf{x}, y)$ for which the loss is positive, and $\langle \cdot \rangle_{\mathbf{x}, y}$ denotes the average over examples and classes ($y = \pm 1$ for binary classification). The dynamical equations on the node parameters simplify too:

$$\begin{cases} \dot{a}_i(t) &= \dfrac{\beta}{\sqrt{d}} \mathbf{w}_i \cdot \langle u(\mathbf{x}, y; t) \theta (\mathbf{w}_i \cdot \mathbf{x}) y \, \mathbf{x} \rangle_{\mathbf{x}, y} \\ \dot{\mathbf{w}}_i(t) &= \dfrac{\beta}{\sqrt{d}} a_i \langle u(\mathbf{x}, y; t) \theta (\mathbf{w}_i \cdot \mathbf{x}) y \, \mathbf{x} \rangle_{\mathbf{x}, y}. \end{cases} \tag{7}$$

Remarkably, the equation on the $\mathbf{w}_i$ is very similar to the one induced by the Hebb rule in biological neural networks.

## 3 Analysis of a linearly separable case

We now focus on a linearly separable model, where the dynamics can be solved explicitly. We consider a reference unit vector $\hat{\mathbf{w}}^*$ in input space and examples distributed according to a spherical probability distribution $P(\mathbf{x})$. We label each example based on the sign of its scalar product with $\hat{\mathbf{w}}^*$, leading to a distribution for $y = \pm 1$: $P(\mathbf{x}, y) = P(\mathbf{x})\theta(y(\hat{\mathbf{w}}^* \cdot \mathbf{x}))$.

In order to be able to explore different training regimes, we adopt a rescaled loss function, similar to the one proposed in Chizat et al. [23]:

$$\mathcal{L}^{\alpha}(\boldsymbol{\theta}) = \frac{1}{\alpha^2 N} \sum_{n=1}^{N} \mathcal{R}\Big[h - \alpha y_n \Big(f(\mathbf{x}_n; \boldsymbol{\theta}) - f(\mathbf{x}_n; \boldsymbol{\theta}_0)\Big)\Big], \tag{8}$$

where $\alpha$ is the rescaling parameter and $\boldsymbol{\theta}_0$ are the parameters at the beginning of training. Subtracting the initial output of the network ensures that no bias is introduced by the specific finite choice of parameters at initialization, while having no influence in the hydrodynamic limit since the output is 0 by construction.

## 3.1 Explicit solution for an infinite training set

We first consider the limit of infinite number of examples, and later discuss the effects induced by a finite training set.

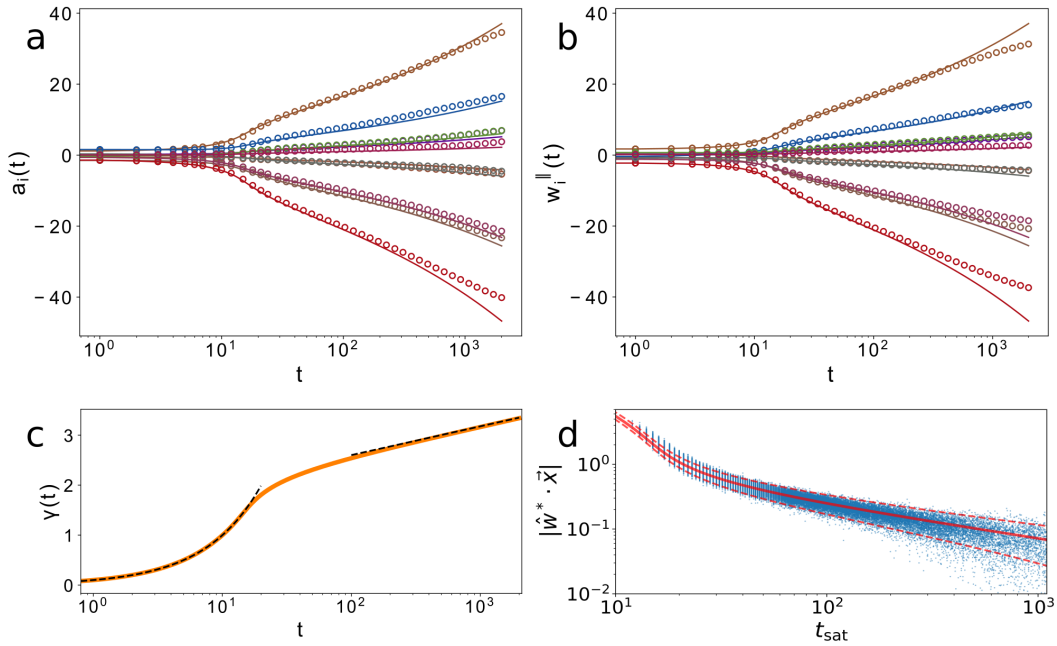

Figure 1: Training of a network with $M = 400$, $N = 10^5$, $d = 100$, $\alpha = 1.0$, $h = 1$, $\beta^* = 10^3$, for $t_{\max} = 2 \cdot 10^3$ timesteps (until all examples are classified) with final generalization error $\sim 0.01$ evaluated on $10^5$ examples. Data and initial parameters are taken from a normal distribution of zero mean and width 1 per dimension. **a, b:** Evolution of ten of the $a_i(t)$s in (a) and of the $w_i^{\parallel}(t)$s in (b) during training (circles) compared to our theoretical prediction (lines) for the same initial values. **c:** Evolution of $\gamma(t)$ obtained through numerical integration of eq. 13 for the parameters of this example. The dashed lines represent the linear approximation near $t = 0$ and the logarithmic slope $\log(t)/4$ for large $\gamma$ (shifted with a fitted constant). **d:** Projection of examples on the vector $\hat{\mathbf{w}}^*$ as a function of the time $t_{\text{sat}}$ when they are first satisfied. The red line is the estimate of our theory, the dashed lines represent our estimate for a standard deviation due to the finite number of nodes $M$ (see Sec. 3.3).

The explicit solution of the training dynamics is obtained making use of the cylindrical symmetry around $\hat{\mathbf{w}}^*$, which implies that the average in the equations of motion (7) does not depend on $\mathbf{w}$, i.e.

$$\langle u(\mathbf{x}, y; t)\theta\,(\mathbf{w} \cdot \mathbf{x})\,y\,\mathbf{x}\rangle_{\mathbf{x},y} = I(t)\hat{\mathbf{w}}^*. \tag{9}$$

where $I(t) \equiv \langle u(\mathbf{x}, y; t)\theta\,(\mathbf{w} \cdot \mathbf{x})\,y\,\mathbf{x} \cdot \hat{\mathbf{w}}^*\rangle_{\mathbf{x},y}$. By plugging the identity (9) into eqs. (6, 7) one finds that the hydrodynamic equation (5) can be solved by the method of the characteristic, where $\rho(\boldsymbol{\theta}, t)$ is obtained by transporting the initial condition through the equations (7). By decomposing the vector

$\mathbf{w}$ in its parallel and perpendicular components with respect to $\hat{\mathbf{w}}^*$, i.e. $\mathbf{w} = w^{\parallel}\hat{\mathbf{w}}^* + \mathbf{w}_{\perp}$, and using the solution $\rho(\boldsymbol{\theta}, t)$, one finds that the parameters $\boldsymbol{\theta}$ at time $t$ are distributed in law as:

$$
\begin{cases}
a(t) & \overset{d}{\sim} & a(0)\cosh(\gamma(t)) + w^{\parallel}(0)\sinh(\gamma(t)) \\
w^{\parallel}(t) & \overset{d}{\sim} & w^{\parallel}(0)\cosh(\gamma(t)) + a(0)\sinh(\gamma(t)) \quad ; \qquad \gamma(t) = \frac{\beta}{\alpha\sqrt{d}}\int_0^t I(t)\mathrm{d}t. \quad (10) \\
\mathbf{w}_{\perp}(t) & \overset{d}{\sim} & \mathbf{w}_{\perp}(0)
\end{cases}
$$

where $a(0), w^{\parallel}(0), \mathbf{w}_{\perp}(0)$ are given by the initial condition distributions: since all initial components of $\mathbf{w}$ were taken as i.i.d. Gaussian, so is $w^{\parallel}(0)$ and every component of $\mathbf{w}_{\perp}(0)$ for any choice of basis. Using the distribution of $\boldsymbol{\theta}$ at time $t$, one can then compute $\langle u(\mathbf{x}, y; t)\theta(\mathbf{w}\cdot\mathbf{x})\, y\, \mathbf{x}\cdot\hat{\mathbf{w}}^*\rangle_{\mathbf{x}, y}$ and hence obtain a self-consistent equation on $I(t)$, which completes the mean-field solution. Similarly, one can obtain explicitly the output function and the indicator function which acquire a simple form:

$$
f(\mathbf{x}; \boldsymbol{\theta}) = \frac{\sinh(2\gamma(t))}{2\sqrt{d}}\hat{\mathbf{w}}^*\cdot\mathbf{x}, \quad (11)
$$

$$
u(\mathbf{x}, y; t) = \theta\left(\frac{2h\sqrt{d}}{\alpha\sinh(2\gamma(t))} - y\hat{\mathbf{w}}^*\cdot\mathbf{x}\right) \quad (12)
$$

where we have used that $f(\mathbf{x}; \boldsymbol{\theta}) = 0$ at $t = 0$. As expected, both functions have cylindrical symmetry around $\hat{\mathbf{w}}^*$. The analytical derivation of these results and the following ones is presented in the SM. Since by definition $I(t) \geq 0$ the function $\gamma(t)$ is monotonously increasing and starts from zero at $t = 0$. To be more specific, we consider two cases: normally distributed data with unit variance in each dimension, and uniform data on the $d$-dimensional unit sphere. The corresponding self-consistent equations on $\gamma(t)$ read respectively:

$$
\dot{\gamma}(t) = \frac{\beta I^N(0)}{\alpha\sqrt{d}}\left(1 - \exp\left[-\frac{2h^2 d}{\alpha^2\sinh^2(2\gamma(t))}\right]\right), \quad (13)
$$

$$
\dot{\gamma}(t) = \frac{\beta I^S(0)}{\alpha\sqrt{d}}\left(1 - \max\left(0, 1 - 4h^2 d/(\alpha^2\sinh^2(2\gamma(t)))\right)^{\frac{d-1}{2}}\right), \quad (14)
$$

where $I^N(0) = 1/\sqrt{2\pi}$ and $I^S(0) = \Gamma\left(\frac{d+2}{2}\right)/(\Gamma\left(\frac{d+1}{2}\right)d\sqrt{\pi})$. Both equations imply that $\gamma(t)\sim t$ for small $t$ and $\gamma(t)\sim\ln t$ for large $t$.

We have now gained a full analytical description of the training dynamics: the node parameters evolve in time following eqs. (10). Note that their trajectory is independent of the training parameters and the initial distribution, which only affect the time dependence, i.e. the "clock" $\gamma(t)$. The change of the output function is given by eq. (11), where one sees that only the amplitude of $f(x, \boldsymbol{\theta})$ varies with time and is governed by $\gamma(t)$. The amplitude increases monotonically so that more examples can be classified above the margin $h$ at later times; the more examples are classified the slower becomes the increase of $\gamma(t)$ and hence the dynamics.

Our theoretical prediction can be directly compared with a simple numerical experiment. Fig. 1 shows the training of a network with $M = 400$ on Gaussian input data. The top panels (**a** and **b**) compare the analytical evolution of the network parameters $a_i$ and $w_i^{\parallel}$ obtained from eqs. (10) to the numerical one. In **c** we plot $\gamma(t)$ (computed numerically) showing that it grows linearly in the beginning and logarithmically at longer times, as expected from theory. In **d** we show a scatter plot illustrating that the time when an example is satisfied is proportional to its projection on the reference vector, following on average our estimate based on eq. (12). Overall, the agreement with the analytical solution is very good. The spread around the analytical solution in panel **d** is a finite-$M$ effect, that we will analyze in Sec. 3.3. The departure from the analytical result (10) happens at large time when the finiteness of the training set starts to matter (the larger is the training set the larger is this time). In fact, for any finite number of examples the empirical average over unsatisfied examples deviates from its population average and the dynamics is modified eventually, and ultimately stops when the whole training set is classified beyond margin. We study this regime in Sec. 3.3.

## 3.2 Lazy learning and rich learning regimes

The presence of the factor $\alpha$ in the loss function (8) allows us to explore explicitly the crossover between different learning regimes, in particular the *"lazy learning"* regime corresponding to

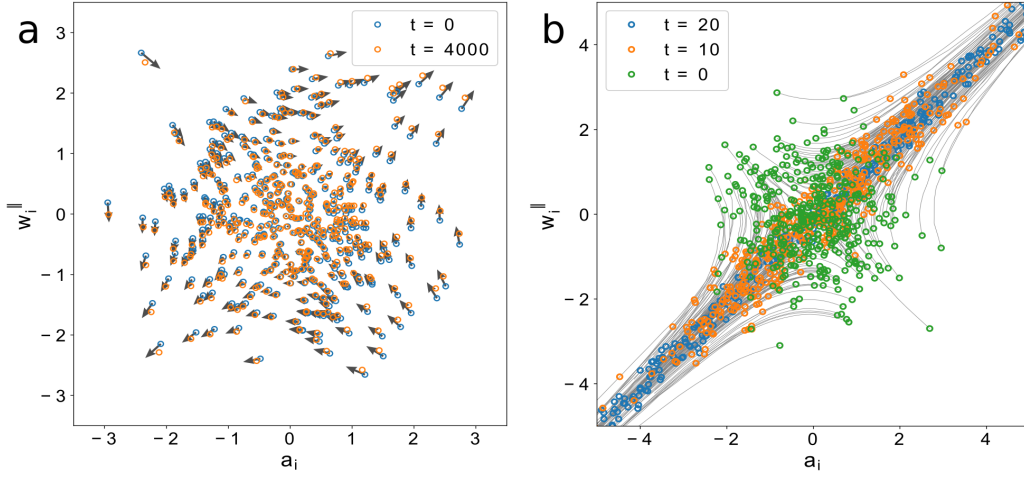

Figure 2: Evolution of $a_i$ and $w_i^{\parallel}$ for a network with $M = 400$, $N = 10^4$, $d = 100$, $h = 1$ in two different regimes. Data and initial parameters are taken from a normal distribution of zero mean and width 1 per dimension. **a:** First and last step of a case with $\alpha = 10^3$ (learning rate $\beta^* = 10^4$, training set is fitted by $t = 3000$, final generalization error $\sim 0.04$). The arrows indicate the analytical derivative at $t = 0$, showing that the evolution is approximately linear. **b:** Initial steps (time indicated in legend) of a case with $\alpha = 10^{-3}$ (learning rate $\beta^* = 1$, training set is fitted by $t = 300$, final generalization error $\sim 0.02$). The gray lines follow the evolution of each node.

$\alpha \to \infty$ [23]. The dynamical equations can be studied in this limit by introducing $\overline{\gamma}(t) = \alpha\gamma(t)$. For concreteness, let us focus on the case of normally distributed data. Taking the $\alpha \to \infty$ limit of eq. (13) one finds the equation for $\overline{\gamma}(t)$:

$$\dot{\overline{\gamma}}(t) = \frac{\beta I^N(0)}{\sqrt{d}} \left( 1 - \exp\left[ -\frac{2h^2 d}{4\overline{\gamma}(t)^2} \right] \right),$$ (15)

As for the evolution of the parameters and the output function, we obtain:

$$\begin{cases} a_i(t) - a_i(0) &= w_i^{\parallel}(0)\dfrac{\overline{\gamma}(t)}{\alpha} + O(\alpha^{-2}) \\ w_i^{\parallel}(t) - w_i^{\parallel}(0) &= a_i(0)\dfrac{\overline{\gamma}(t)}{\alpha} + O(\alpha^{-2}) \end{cases} \quad ; \quad \alpha f(\mathbf{x};\boldsymbol{\theta}) = \frac{\overline{\gamma}(t)}{\sqrt{d}}\hat{\mathbf{w}}^* \cdot \mathbf{x}.$$ (16)

The equations above provide an explicit solution of lazy learning dynamics and illustrate its main features: the $\boldsymbol{\theta}_i$ evolves very little and along a fixed direction, in this case given by $(w_i^{\parallel}(0), a_i(0), 0)$. Despite the small changes in the nodes parameters, of the order of $1/\alpha$, the network does learn since classification is performed through $\alpha f(\mathbf{x};\boldsymbol{\theta})$ which has an order one change even for $\alpha \to \infty$. In this regime, the correlation between $a$ and $w^{\parallel}$ only increases slightly, but this is enough for classification, since an infinite amount of displacements in the right direction is sufficient to solve the problem. On the contrary, when $\alpha$ is of order one or smaller, the dynamics is in the so-called *"rich learning"* regime [27]. At the beginning of learning, the initial evolution of the $\boldsymbol{\theta}_i$s follows the same linear trajectories of the lazy-learning regime. However, at later stages, the trajectories are no more linear and the norm of the weights increases exponentially in $\gamma(t)$, stopping only at very large values of $\gamma$ when all nodes are almost aligned with $\hat{\mathbf{w}}^*$ (for small $\alpha$). Note that, as observed in Geiger et al. [34], with the standard normalization $1/\sqrt{M}$ it would be the parameter $\alpha\sqrt{M}$ governing the crossover between the two regimes.

We compare the two dynamical evolutions in Fig. 2. The left panel (**a**) shows the displacement of parameters between initialization and full classification (zero training loss) for a network with $\alpha = 10^3$. As expected, the displacement is small and linear. A very different evolution takes place for $\alpha = 10^{-3}$ in the right (**b**) panel. The trajectories are non-linear, and all nodes approach large values close to the $a = w^{\parallel}$ line at the end of the training. Correspondingly, the initially isotropic Gaussian

distribution evolves towards one with covariance matrix $\cosh(2\gamma)$ on the diagonal and $\sinh(2\gamma)$ off diagonal.

Note that for all values of $\alpha$, even very large ones, the trajectories of the $\boldsymbol{\theta}_i$s are identical and given by eqs. (10). What differs is the "clock" $\gamma(t)$, in particular for large $\alpha$ the system remains for a much longer time in the lazy regime. This is true as long as the number of training samples is infinite. Instead, if the number of data is finite, the dynamics stops once the whole training set is fitted: for large $\alpha$ this happens before the system is able to leave the lazy regime, whereas for small $\alpha$ a full non-linear (rich) evolution takes place. Hence, the finiteness of the training set leads to very distinct dynamics and profoundly different "trained" models (having both fitted the training dataset) with possibly different generalization properties [25, 34, 35].

### 3.3 Beyond mean-field theory

The solution we presented in the previous sections holds in the limit of an infinite number of nodes and of training data. Here we study the corrections to this asymptotic limit, and discuss the new phenomena that they bring about.

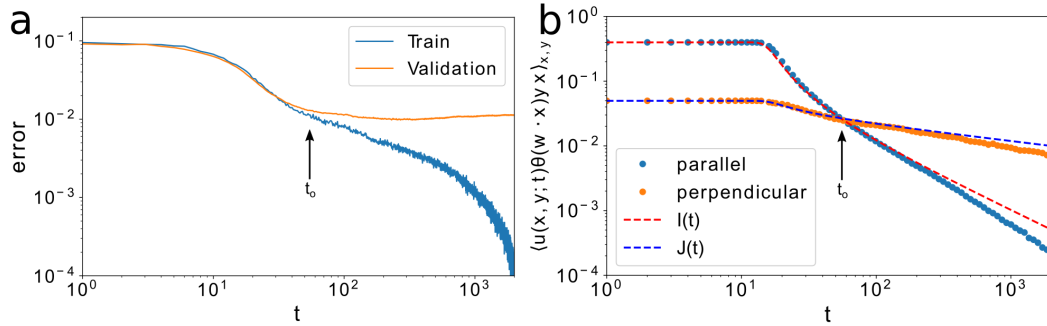

Figure 3: **a:** Training (blue) and generalization (orange) error (fraction of misclassified examples), during training with same parameters as Fig. 1. **b:** Components of $\langle u(\mathbf{x}, y; t)\theta (\mathbf{w} \cdot \mathbf{x}) \, y \, \mathbf{x}\rangle_{\mathbf{x},y}$ along $\hat{\mathbf{w}}^*$ (parallel) and perpendicular to it, during training. The dots are numerical results for the same training show in **a**. The lines represent our analytical predictions $I(t)$ and $J(t)/\sqrt{N}$ for the same parameters.

**Finite number of nodes.** In the large $M$ limit the $a_i$ and $\mathbf{w}_i$ are Gaussian i.i.d. variables. By the central limit theorem, the function (2) concentrates around its average, and has negligible fluctuations of the order of $1/\sqrt{M}$ when $M \to \infty$. If $M$ is large but finite (keeping an infinite training set), these fluctuations of $f(x, \boldsymbol{\theta})$ are responsible for the leading corrections to mean-field theory. In the SM we compute explicitly the variance of the output function, $\lim_{M \to \infty} M\mathrm{Var}[f(x, \boldsymbol{\theta})] = \sigma_f^2(t)$, with

$$\sigma_f^2(t) \equiv ((5\cosh^2(2\gamma(t)) - 2\cosh(2\gamma(t)) - 3)(\hat{\mathbf{w}}^* \cdot \mathbf{x})^2 + 2\cosh(2\gamma(t))\,|\mathbf{x}|^2)/(4d) \quad (17)$$

The main effect of this correction is to induce a spread in the dynamics, e.g. of the data with same satisfaction time. This phenomenon is shown in Fig. 1(**d**) for $M = 400$, where we compare the numerical spread to an estimate of the values of $\hat{\mathbf{w}}^* \cdot \mathbf{x}$ such that the hinge is equal to the average plus or minus one standard deviation (details on this estimate in the SM).

**Finite number of data.** We now consider a finite but large number of examples $N$ (keeping infinite the number of nodes). In the large $N$ limit the empirical average over the data in $\langle u(\mathbf{x}, y; t)\theta (\mathbf{w} \cdot \mathbf{x}) \, y \, \mathbf{x}\rangle_{\mathbf{x},y}$ converges to its mean $I(t)\hat{\mathbf{w}}^*$. The main effect of considering a finite $N$ is that the empirical average fluctuates around this value. Using the central limit theorem we show in the SM that the leading correction to the asymptotic result reads:

$$\langle u(\mathbf{x}, y; t)\theta (\mathbf{w} \cdot \mathbf{x}) \, y \, \mathbf{x}\rangle_{\mathbf{x},y} = I(t)\hat{\mathbf{w}}^* + \frac{J(t)}{\sqrt{N}}\delta\mathbf{w}_\perp + O(1/N) \quad (18)$$

where $\delta\mathbf{w}_\perp$ is a unitary random vector perpendicular to $\hat{\mathbf{w}}^*$ and $J(t) \equiv \sqrt{(d-1)f^U(t)/2}$. The term $f^U(t) \equiv \langle u(\mathbf{x}, y; t)\rangle_{\mathbf{x},y}$, the fraction of unsatisfied examples at time $t$, controls the strength of the

correction, as expected since only unsatisfied data contribute to the empirical average $\langle\cdot\rangle_{\mathbf{x},y}$. The vector on the RHS of (18) is the one towards which all the $\mathbf{w}_i$ align, see eqs. (10). Therefore, the main effect of the correction (18) is for the nodes parameters to align along a direction which is slightly different from $\hat{\mathbf{w}}^*$ and dependent on the training set. This naturally induces different accuracies between the training and the test sets, i.e. it leads to *overfitting*.[2] Note that the strength of the signal, $I(t)$, is roughly of the order of the fraction of unsatisfied data $f^U(t)$, whereas the noise due to the finite training set is proportional to the square root of it. The larger time, the smaller $f^U(t)$ is, hence the stronger are the fluctuations with respect to the signal. In Fig. 3(**b**) we compute numerically the components of $\langle u(\mathbf{x}, y; t)\theta\left(\mathbf{w}\cdot\mathbf{x}\right)y\,\mathbf{x}\rangle_{\mathbf{x},y}$ parallel and perpendicular to $\hat{\mathbf{w}}^*$, and compare them to $I(t)$ and $J(t)/\sqrt{N}$. Remarkably, we find a very good agreement even for times when $J(t)/\sqrt{N}$ is no longer a small correction. This suggest that an estimate of the time $t_o$ when overfitting takes place is given by $I(t_o) = J(t_o)/\sqrt{N}$. We test this conjecture in panel (**a**): indeed the two contributions are of the same order of magnitude for $t_o \sim 50$, which is around the time when training and validation errors diverge.

## 3.4 Mislabeling

We now briefly address the effects due to noise in the labels, see the SM for detailed results and numerical experiments. Mislabeling is introduced by flipping the label of a small fraction $\delta$ of the examples. The main effect is to decrease the strength of the signal, $I(t)$, since the mislabeled data lead to an opposite contribution in (9) with respect to the correctly labeled ones. In the asymptotic limit of infinite $N$ and $M$, the reduction of the signal slows down the dynamics, which stops when the number of unsatisfied correct examples equals the one of mislabeled ones. For large but finite $N$, the noise $J(t)/\sqrt{N}$ is enhanced with respect to the signal because its strength is related to the fraction of *all* unsatisfied examples, and not just the correctly labeled ones. Hence, overfitting is stronger and takes place earlier with respect to the case analyzed before.

## 4 Discussion and Experiment

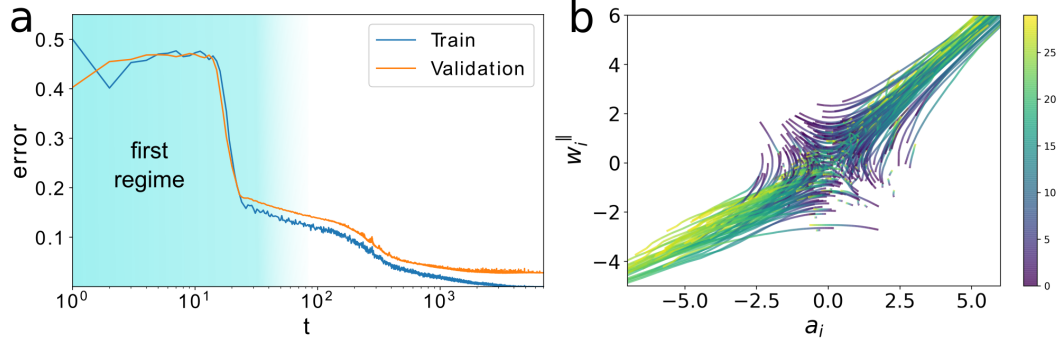

Figure 4: **a:** Training (blue) and generalization (orange) error for a network with $M = 400$, trained on $N = 10^4$ MNIST data ($d = 784$), with parity labels. Inputs are only rescaled by a factor $1/255$, no further processing is done. The training is performed with $\beta^* = 1000$, $\alpha = 1$, $h = 1$ and the validation error on $10^4$ examples is $\sim 0.03$ after 2000 evolution steps. The shaded area represents the area where our theory applies. **b:** Evolution of $a_i$ and $w_i^{\parallel}$ in the first 30 steps of training. The color (see color bar) represents the step of evolution.

We have provided an analytical theory for the dynamics of a single hidden layer neural network trained for binary classification with linear hinge loss. We have found two dynamical regimes: a first one, correctly accounted for by mean-field theory, in which every node has its own dynamics with a time-dependent dataset determined self-consistently from the average nodes population. During this evolution the nodes parameters align with the direction of the reference classification vector. In the second regime, which is not accounted for by mean-field theory, the noise due to the finite

training set becomes important and overfitting takes place. The merit of the model we focused on is that, thanks to its simplicity, several effects happening in real networks can be studied in detail analytically. Several works have shown distinct dynamical regimes in the training dynamics: first the network learns coarse grained properties, later on it captures the finer structure, and eventually it overfits [8, 13, 36, 37]. Given the simplicity of the dataset considered, we expect our model to describe the first regime but not the second one, which would need a more complex model of data.

In particular, the effective one-dimensional nature of the $\mathbf{w}$ evolution is due to the cylindrical symmetry of the data, resulting in a direction-independent expression for the integral in eq.(9). In a more general setting, we can still expect to recover a similar behavior at the beginning of training, where the difference between the two classes averages dominates most of the dynamics. After that, the integral will depend more and more on the direction of $\mathbf{w}$, leading to specialization and a departure from our simple model. To test this conjecture, we train our network to classify the parity of MNIST handwritten digits [38]. To establish a relationship with our case, we define $\hat{\mathbf{w}}^*$ as the direction of the difference between the averages of the two parity sets. We can now define $w^\parallel$ for each node, and study the dynamics of $a_i, w_i^\parallel$. We report in Fig. 4 the evolution of these parameters in the early steps of training, in which the training loss decreases of $65\%$ of its initial value (Fig. 4**a**). The evolution of the parameters (Fig. 4**b**) bears a strong resemblance with our findings, see the remarkable similarity with Fig. 2(**b**). A similar experiment for even richer datasets (CIFAR10 and ImageNet) is presented in the SM.

## Broader Impact

Given the purely theoretical scope of this paper, it does not seem to present any foreseeable societal consequence.

## Acknowledgments and Disclosure of Funding

We thank S. d'Ascoli and L. Sagun for discussions, and M. Wyart for exchanges about his work on a similar model [39].

We acknowledge funding from the French government under management of Agence Nationale de la Recherche as part of the "Investissements d'avenir" program, reference ANR-19-P3IA-0001 (PRAIRIE 3IA Institute) and from the Simons Foundation collaboration "Cracking the Glass Problem" (No. 454935 to G. Biroli).

## Footnotes

[1]In the NTK (or "lazy") limit [23–25] general losses have been considered.

[2]The two accuracies instead coincide for $N \to \infty$, since all possible data are seen during the training and no overfitting is present in the asymptotic limit.

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
