[Supplementary Material]

# An analytic theory of shallow networks dynamics for hinge loss classification — Supplementary Material

**Franco Pellegrini**
Laboratoire de Physique de l'École normale supérieure, ENS,
Université PSL, CNRS, Sorbonne Université, Université de Paris
F-75005 Paris, France

**Giulio Biroli**
Laboratoire de Physique de l'École normale supérieure, ENS,
Université PSL, CNRS, Sorbonne Université, Université de Paris
F-75005 Paris, France

## 1  Explicit calculations

### 1.1  Derivation of the hydrodynamics mean-field equation

In order to simplify the derivation in the following we use a compact notation for the function $f$:

$$f(\mathbf{x}; \boldsymbol{\theta}) = \frac{1}{M} \sum_{i=1}^{M} \overline{\sigma}(\mathbf{x}; \boldsymbol{\theta}_i) \tag{1}$$

where $\overline{\sigma}(\mathbf{x}; \boldsymbol{\theta}_i) \equiv a_i \sigma\left(\frac{\mathbf{w}_i \cdot \mathbf{x}}{\sqrt{d}}\right)$, and for the gradient flow equations on the parameters of the network:

$$\dot{\boldsymbol{\theta}}_i = -\frac{\beta}{N} \sum_{n=1}^{N} \frac{\partial \ell(y_n, f(\mathbf{x}_n; \boldsymbol{\theta}))}{\partial f} \frac{\partial \overline{\sigma}(\mathbf{x}_n; \boldsymbol{\theta}_i)}{\partial \boldsymbol{\theta}_i}. \tag{2}$$

The strategy to derive hydrodynamics mean-field equations developed in physics consists in using the following equation, valid for $M \to \infty$ and any test function $H$:

$$\frac{1}{M} \sum_{i=1}^{M} H(\boldsymbol{\theta}_i(t)) = \int d\boldsymbol{\theta} H(\boldsymbol{\theta}) \rho(\boldsymbol{\theta}, t) \tag{3}$$

and then in differentiating RHS and LHS with respect to time, see e.g. [1]. The important point here (and later) is that the density $\rho(\boldsymbol{\theta}, t)$, which depends on the random initial conditions, concentrates in the large $M$ limit due to the nature of the interaction between parameters, which is only through the function $f$, and the type of distributions considered for the initial conditions.[1] The derivative of the RHS leads to

$$\int d\boldsymbol{\theta} H(\boldsymbol{\theta}) \partial_t \rho(\boldsymbol{\theta}, t) \tag{4}$$

whereas the derivative of the LHS reads:

$$-\frac{\beta}{M} \sum_{i=1}^{M} \nabla_{\boldsymbol{\theta}} H(\boldsymbol{\theta}_i(t)) \frac{1}{N} \sum_{n=1}^{N} \frac{\partial \ell(y_n, f(\mathbf{x}_n; \boldsymbol{\theta}))}{\partial f} \nabla_{\boldsymbol{\theta}} \overline{\sigma}(\mathbf{x}_n; \boldsymbol{\theta}_i(t)). \tag{5}$$

We now use the identity:

$$\frac{\delta\mathcal{L}[\rho(\boldsymbol{\theta},t)]}{\delta\rho(\boldsymbol{\theta},t)} = \frac{1}{N}\sum_{n=1}^{N}\frac{\partial\ell(y_n, f(\mathbf{x}_n;\boldsymbol{\theta}))}{\partial f}\overline{\sigma}(\mathbf{x}_n;\boldsymbol{\theta}(t)) \tag{6}$$

to rewrite the LHS as

$$-\frac{\beta}{M}\sum_{i=1}^{M}\nabla_{\boldsymbol{\theta}}H(\boldsymbol{\theta}_i(t))\nabla_{\boldsymbol{\theta}_i(t)}\left.\frac{\delta\mathcal{L}[\rho(\boldsymbol{\theta},t)]}{\delta\rho(\boldsymbol{\theta},t)}\right|_{\boldsymbol{\theta}=\boldsymbol{\theta}_i(t)}. \tag{7}$$

For $M \to \infty$ this expression can be rewritten as

$$-\beta\int d\boldsymbol{\theta}\rho(\boldsymbol{\theta},t)\nabla_{\boldsymbol{\theta}}H(\boldsymbol{\theta})\nabla_{\boldsymbol{\theta}}\frac{\delta\mathcal{L}[\rho(\boldsymbol{\theta},t)]}{\delta\rho(\boldsymbol{\theta},t)} = \int d\boldsymbol{\theta}H(\boldsymbol{\theta})\left[\beta\nabla_{\boldsymbol{\theta}}\left(\rho(\boldsymbol{\theta},t)\nabla_{\boldsymbol{\theta}}\frac{\delta\mathcal{L}[\rho(\boldsymbol{\theta},t)]}{\delta\rho(\boldsymbol{\theta},t)}\right)\right] \tag{8}$$

where we have used an integration by part to obtain the last identity. Since the expressions in (4) and (8) are equal for any test function $H$, we obtain that the density $\rho(\boldsymbol{\theta},t)$ verifies the equation written in the main text:

$$\partial_t\rho(\boldsymbol{\theta},t) = \beta\nabla_{\boldsymbol{\theta}}\left(\rho(\boldsymbol{\theta},t)\nabla_{\boldsymbol{\theta}}\frac{\delta\mathcal{L}[\rho(\boldsymbol{\theta},t)]}{\delta\rho(\boldsymbol{\theta},t)}\right) \quad, \quad \rho(\boldsymbol{\theta},0) = \mathcal{N}(0,\mathbb{I}). \tag{9}$$

The initial condition for $\rho(\boldsymbol{\theta},t)$ is a Gaussian distribution since the parameters at initialization are i.i.d. Gaussian variables.

## 1.2   Calculation of $I(t)$

We want to compute the integral of Eq. (9) of the main text:

$$\langle u(\mathbf{x},y;t)\theta\left(\mathbf{w}\cdot\mathbf{x}\right)y\mathbf{x}\rangle_{\mathbf{x},y} = \int\sum_{y=\pm1}u(\mathbf{x},y;t)\theta\left(\mathbf{w}\cdot\mathbf{x}\right)y\mathbf{x}P(\mathbf{x},y)\mathrm{d}\mathbf{x} \tag{10}$$

for the task and distributions mentioned in the text.

Let us start by observing that since $P(\mathbf{x},y)$ has spherical symmetry and $u(\mathbf{x},y;t)$ has cylindrical symmetry around $\hat{\mathbf{w}}^*$ and is symmetric under inversion along $\hat{\mathbf{w}}^*$ (because of the label symmetry of the problem), the whole integrand without the $\theta\left(\mathbf{w}\cdot\mathbf{x}\right)$ is symmetric under inversion operation. Indeed, $P(\mathbf{x},y) = P(-\mathbf{x},-y)$, $u(\mathbf{x},y;t) = u(-\mathbf{x},-y;t)$ and $y\mathbf{x} = \mathrm{sign}(\hat{\mathbf{w}}^*\cdot\mathbf{x})\mathbf{x} = \mathrm{sign}(\hat{\mathbf{w}}^*\cdot(-\mathbf{x}))(-\mathbf{x})$. The effect of the $\theta\left(\mathbf{w}\cdot\mathbf{x}\right)$ term is to select one particular half-space over which the integral is done. However, because of the symmetric under inversion the integral on *any* half space is equivalent, hence the result is independent of $\mathbf{w}$. Moreover for any direction orthogonal to $\hat{\mathbf{w}}^*$, the integrand is odd under inversion of that component, and is therefore 0. The only component different from zero is then the one along $\hat{\mathbf{w}}^*$, dubbed $I(t)$ in the text. Let us define $\hat{\mathbf{w}}^*\cdot\mathbf{x} = x^{\|}$ and notice that that $yx^{\|} = \mathrm{sign}(x^{\|})x^{\|} = |x^{\|}|$ so that we can for simplicity consider the integral on the positive values

$$I(t) = \int_{x^{\|}>0}\sum_{y=\pm1}u(\mathbf{x},y;t)x^{\|}P(\mathbf{x},y)\mathrm{d}\mathbf{x}. \tag{11}$$

We will now consider the specific expression found in the main text $u(\mathbf{x},y;t) = \theta(H - yx^{\|})$, and for the noiseless case $P(\mathbf{x},y) = P(\mathbf{x})\theta(yx^{\|})$.

In the case of normally distributed data, all orthogonal directions integrate to 1 and we are left with a simple Gaussian integral

$$I(t) = \int_0^H x^{\|}\mathcal{N}_{0,1}(x^{\|})\mathrm{d}x^{\|} = \frac{1}{\sqrt{2\pi}}\left(1 - e^{-H^2/2}\right). \tag{12}$$

With $H = \frac{2h\sqrt{d}}{\alpha\sinh(2\gamma(t))}$ and $\dot{\gamma}(t) = \frac{\beta}{\alpha\sqrt{d}}I(t)$, we recover Eq. (13) from the text.

For the case of data uniformly distributed on the $d-1$-dimensional unit sphere in $d$ dimensions, we divide by the sphere surface $S_{d-1}$ and integrate on the $d-1$ angular coordinates. Because of the symmetry, we perform $d-2$ angular integrals and obtain the surface of the $d-2$-dimensional sphere.

The $u(\mathbf{x}, y; t) = \theta(H - yx^{\|})$ limit will set the extreme of integration to $\arccos(H)$ for $H < 1$ and not affect the integral otherwise. Considering for simplicity directly the $H < 1$ limit we obtain:

$$I(t) = \frac{S_{d-2}}{S_{d-1}} \int_{\arccos H}^{\pi/2} \cos(\phi) \sin^{d-2}(\phi) \mathrm{d}\phi = \frac{S_{d-2}}{(d-1)S_{d-1}} \left[ 1 - (\sin \arccos(H))^{d-1} \right]. \quad (13)$$

Using the equation $S_{n-1} = n\pi^{n/2}/\Gamma(n/2 + 1)$ for the sphere surface and properly accounting for the different $H$ cases we recover Eq. (14) from the text.

## 1.3 Calculation of finite size quantities

**Finite number of nodes.** To estimate the fluctuations due to a finite number of nodes, we will have to estimate the width of the output distribution for a given set of parameters. Let us explicit from equations (10) of the main text for the parameters evolution that, starting from i.i.d. Gaussian initialization, the distribution of $(a, w^{\|})$ is

$$\rho(a(t), w^{\|}(t)) = \mathcal{N}\left(0, \begin{pmatrix} \cosh(2\gamma(t)) & \sinh(2\gamma(t)) \\ \sinh(2\gamma(t)) & \cosh(2\gamma(t)) \end{pmatrix}\right), \quad (14)$$

while all perpendicular components remain i.i.d.

The average output $f(\mathbf{x}; \boldsymbol{\theta})$ for an example $\mathbf{x}$ can then be simply computed from its definition as

$$\int_{-\infty}^{\infty} \mathrm{d}a(t) \int_{0}^{\infty} \mathrm{d}w^{\|}(t) \frac{a(t)w^{\|}(t)x^{\|}}{\sqrt{d}} \rho(a(t), w^{\|}(t)) = \frac{x^{\|}}{2\sqrt{d}} \left\langle a(t)w^{\|}(t) \right\rangle = \frac{\sinh(2\gamma(t))x^{\|}}{2\sqrt{d}} \quad (15)$$

(all orthogonal integrals being equal to 1), having defined again $\hat{\mathbf{w}}^* \cdot \mathbf{x} = x^{\|}$. This proves Eq. (11) of the main text.

In order to estimate the fluctuations we should however compute the integral (we drop the t dependence for simplicity)

$$\left\langle f(\mathbf{x}; \boldsymbol{\theta})^2 \right\rangle_{\boldsymbol{\theta}} = \frac{1}{d} \int \mathrm{d}a\, \mathrm{d}\mathbf{w}\, a^2 (\mathbf{w} \cdot \mathbf{x})^2\, \theta(\mathbf{w} \cdot \mathbf{x}) \rho(a, \mathbf{w}). \quad (16)$$

Since the integral is 1 for any direction perpendicular to $\mathbf{x}$, this is more easily done considering the distribution of $w_x = \mathbf{w} \cdot \hat{\mathbf{x}}$ (with $\hat{\mathbf{x}} = \mathbf{x}/|\mathbf{x}|$). Defining $\hat{\mathbf{w}}^+$ as $(\mathbf{x} - x^{\|}\hat{\mathbf{w}}^*)/|\mathbf{x} - x^{\|}\hat{\mathbf{w}}^*|$, i.e. the versor in the direction of $\hat{\mathbf{x}}$ perpendicular to $\hat{\mathbf{w}}^*$, we can write $\hat{\mathbf{x}} = \cos\theta \hat{\mathbf{w}}^* + \sin\theta \hat{\mathbf{w}}^+$ and calling $w^+ = \mathbf{w} \cdot \hat{\mathbf{w}}^+$ (being a component perpendicular to $\hat{\mathbf{w}}^*$ and therefore i.i.d) we can explicit $w_x = w^{\|} \cos\theta + w^+ \sin\theta$.

We can thus write the distribution for this component as

$$\rho(a(t), w_x(t)) = \mathcal{N}\left(0, \begin{pmatrix} \cosh(2\gamma(t)) & \sinh(2\gamma(t))\cos\theta \\ \sinh(2\gamma(t))\cos\theta & \cosh(2\gamma(t))\cos^2\theta + \sin^2\theta \end{pmatrix}\right), \quad (17)$$

and the integral as just

$$\frac{|\mathbf{x}|^2}{d} \int \mathrm{d}a\, \mathrm{d}w_x\, a^2 w_x^2\, \theta(w_x) \rho(a, w_x) =$$
$$= \frac{|\mathbf{x}|^2}{2d} \left( \cosh^2(2\gamma(t))\cos^2\theta + \cosh(2\gamma(t))\sin^2\theta + 2\sinh^2(2\gamma(t))\cos^2\theta \right). \quad (18)$$

The total spread due to this is thus

$$\sigma_f^2(t) \equiv \left\langle f(\mathbf{x}; \boldsymbol{\theta})^2 \right\rangle_{\boldsymbol{\theta}} - \left\langle f(\mathbf{x}; \boldsymbol{\theta}) \right\rangle_{\boldsymbol{\theta}}^2 =$$
$$= \frac{|\mathbf{x}|^2}{4d} \left[ \left( (5\cosh^2(2\gamma(t)) - 2\cosh(2\gamma(t)) - 3) \cos^2\theta + 2\cosh(2\gamma(t)) \right] \right], \quad (19)$$

which is equivalent to Eq. (17) in the main text.

To estimate the error in Fig. 1d of the main text, we ask what are the values of $x^{\|} = \mathbf{x}\cos\theta$ such that the average output plus or minus a standard deviation, divided by $\sqrt{M}$, would be equal to the threshold. Since the standard deviation involves $|\mathbf{x}|^2$, we estimate its average value for points

with a given $x^\|$, i.e. $\left\langle |\mathbf{x}|^2 \,|_{x^\|} \right\rangle = x^{\|\,2} + d - 1$. The variance is thus the sum of two terms: $\sigma_\|^2 = \left( \left(5\cosh^2(2\gamma(t)) - 3\right)/(4dM) \right)$ multiplying $x^{\|\,2}$ and a constant $\sigma_0^2 = (d-1)\cosh(2\gamma(t))/(2dM)$. Requesting that $h/\alpha = \sinh(2\gamma(t))x^\|_\pm/(2\sqrt{d}) \pm \sqrt{\sigma_\|^2 x^{\|\,2}_\pm + \sigma_0^2}$ we find:

$$x^\|_\pm = \frac{1}{\sinh^2(2\gamma(t))/(4d) - \sigma_\|^2} \left[ \frac{h\sinh(2\gamma(t))}{2\alpha\sqrt{d}} \pm \sqrt{\frac{h\sigma_\|^2}{\alpha^2} + \frac{\sigma_0^2 \sinh^2(2\gamma(t))}{4d} - \sigma_0^2\sigma_\|^2} \right]. \quad (20)$$

These values are the dashed lines reported in Fig.1d of the main text.

Figure 1: Evolution of $|\mathbf{w}_\perp(t)|$ for the same evolution as Fig. 1 of the main text.

**Finite number of data.** To estimate the fluctuations due to finite number of data in $\langle u(\mathbf{x},y;t)\theta\left(\mathbf{w}\cdot\mathbf{x}\right)y\,\mathbf{x}\rangle_{\mathbf{x},y}$ in the direction perpendicular to $\hat{\mathbf{w}}^*$, we use the central limit theorem, which gives fluctuations of the order $\left\langle \left(u(\mathbf{x},y;t)\theta\left(\mathbf{w}\cdot\mathbf{x}\right)y\,\mathbf{x}\right)^2 \right\rangle_{\mathbf{x},y}/N$. We refer to section 1.2 for the general symmetry considerations about that integral: in the case of normally distributed data, and if all data are not satisfied, i.e. $u(\mathbf{x},y;t) = 1$ inside the empirical average over data, then for any given direction orthogonal to $\hat{\mathbf{w}}^*$ one obtains $1/2N$. Since there are $d-1$ such direction, this means that that considering finite number of data leads to a fluctuating component orthogonal to $\hat{\mathbf{w}}^*$ of norm of the order of $\sqrt{(d-1)/(2N)}$.

Let us consider now the case in which only $N^U = f^U N$ examples remain to satisfy, then the number of terms in the empirical sum is $N^U$ instead of $N$. In consequence, we obtain the same results than previously for the variance, but with an extra-factor $f^U$ in front, thus leading to an error of order $\sqrt{(d-1)f^U/(2N)} \equiv J(t)/\sqrt{N}$.

Estimating $f^U(t)$ for normally distributed data, and with the specific expression $u(\mathbf{x},y;t) = \theta(H - yx^\|)$ is then a simple Gaussian integral:

$$f^U(t) \equiv \langle u(\mathbf{x},y;t)\rangle_{\mathbf{x},y} = \int_{-H}^{H} \mathcal{N}_{0,1}(x^\|)\mathrm{d}x^\| = \mathrm{erf}\left(\frac{H}{\sqrt{2}}\right). \quad (21)$$

Computing this for normally distributed data leads to:

$$J(t) = \sqrt{\frac{(d-1)}{2}\mathrm{erf}\left(\frac{h\sqrt{2d}}{\alpha\sinh(2\gamma(t))}\right)}, \quad (22)$$

as was used to compute the estimates in Fig.3b in the main text.

Figure 2: **a:** Training (blue) and generalization (orange) error (fraction of misclassified examples), during a training with a small fraction $\delta = 0.01$ of mislabeled examples. Training parameters: $M = 400$, $N = 10^5$, $d = 100$, $\alpha = 1.0$, $\beta^* = 10^3$, timesteps $t_{\max} = 10^4$, validation on $10^5$ examples. **b:** Components of $\langle u(\mathbf{x}, y; t)\theta(\mathbf{w} \cdot \mathbf{x}) y \, \mathbf{x} \rangle_{\mathbf{x},y}$ along $\hat{\mathbf{w}}^*$ (parallel) and perpendicular to it, during training. The dots are numerical results for the same training show in **a**. The lines represent our analytical predictions $I_\delta(t)$ and $J_\delta(t)$ for the same parameters (Eqs. (23) and (24)). **c, d:** Evolution of a sample (10) of the $a_i(t)$ (c) and $w_i^{\parallel}(t)$ (d) during training (circles) compared to our theoretical prediction (lines) for the noiseless case with the same initial values and parameters. **e:** Evolution of $|\mathbf{w}_\perp(t)|$ for the same sample of nodes.

## 1.4 Calculations for the mislabeling case

We now analyze the case, qualitatively described in the text, where a small fraction $\delta$ of the examples has been mislabeled as belonging to the opposite class.

Looking back at Eq. (11) and with $u(\mathbf{x}, y; t) = \theta(H - yx^{\parallel})$, it is clear that with an infinite number of examples the mislabeled ones are simply never classified, so that the $1 - \delta$ fraction of correct examples gives rise to a normal dynamics, while the $\delta$ fraction of opposite ones contributes an opposite term of constant magnitude. The effective integrals entering the dynamics are thus in this case

$$I_\delta(t) = (1 - \delta)I(t) - \delta I(0), \tag{23}$$

and would drive the dynamics until the two contributions are equal.

When considering a finite number of data, as discussed in Sec. 1.3, the number of unsatisfied examples with the correct label amounts to $(1 - \delta)f^U(t)$, but since all the mislabeled examples are unsatisfied the total number will be incremented by $\delta$ leading to $f_\delta^U(t) = (1 - \delta)f^U(t) + \delta$.

Again, evaluating this for the normally distributed case we find:

$$J_\delta(t) = \sqrt{\frac{(d - 1)}{2}\left[(1 - \delta)\mathrm{erf}\left(\frac{\sqrt{2d}}{\alpha \sinh(2\gamma(t))}\right) + \delta\right]}. \tag{24}$$

# 2 Further numerical experiments

## 2.1 Evolution of $\mathbf{w}_\perp$

We report in Fig. 1 the perpendicular component of the weights for a selection of nodes for the same example shown in Fig. 1 of the main text. As expected, the perpendicular component does not evolve for most of the training, and only increases moderately when we move into the overfitting regime.

## 2.2  Quantities for the mislabeling case

We report here in Fig. 2 some of the same quantities shown in Fig. 1 and Fig. 3 of the main text, for a case where a small fraction $\delta = 0.01$ of the examples are mislabeled. As discussed in the main text, we can see how the dynamics still follows our estimate initially, then diverges into a much stronger overfitting state. Panel **b** shows a comparison of numerical quantities to our estimates of Sec. 1.4: our estimate are still accurate up to the overfitting regime, after which the dynamics changes qualitatively.

## 2.3  Experiments on CIFAR-10 and ImageNet

Figure 3: Training and validation error (left) and parameter evolution in the $(a, w^{\parallel})$ plane for the early stages of evolution (right) for data from CIFAR-10 (top) and ImageNet (bottom). The dashed lines on the left indicate the cutoff steps considered in the corresponding plots on the right.

We present here in Fig. 3 the results of a brief training of our architecture on data from CIFAR-10 [4] and ImageNet [5].

For CIFAR-10, we consider the input $32 \times 32$ color images as vectors of size 3072, taking a subset of $10^4$ images for both training and validation. We rescale each dimension by a factor $1/255$ and remove the average over the training set, to avoid a common evolution of all classes. Since the considered hinge loss requires 2 classes, we divide the original 10 classes in 2 random groups (based on parity of the numeric label). We can then take the reference vector $\hat{\mathbf{w}}^*$ as the difference between the averages over the two classes. We then proceed to train a network with $M = 500$ hidden nodes until the whole training set is learnt (top left in Fig. 3), obtaining an unremarkable final validation error of $\sim 28\%$, as expected for such a simple architecture on a rather challenging task. If we however focus on the early evolution of the weights $a$ and $w^{\parallel}$ (top right in Fig. 3), we can see how up to step $\sim 200$ the shape of the evolution is still similar to what was predicted by our simple model (see Fig. 2 in the main text). After this, the richer shape of the dataset leads to a more complex evolution, not shown here.

In the case of ImageNet data, since we require a fixed-size input we consider a fixed resolution $32 \times 32$ version of the images, leading to an input size equal to CIFAR-10. The same rescaling by a factor $1/255$ and removing of the average over the training set is also applied. However, even so simplified, we verified that a random binary division of all 1000 ImageNet classes would be an almost impossible task for our simple architecture, leading to validation errors no better than random. We suspect this to be due to the rich variety of input and great similarity of many classes (e.g. different dog races). To circumvent this problem, we exploit the fact that the first 398 classes are from the

animal kingdom (i.e. possibly more homogeneous) and verify that a split between the first/last 500 classes represent a slightly more feasible task. As in the other cases, we take $\hat{\mathbf{w}}^*$ as the difference between the averages over the two classes. We further push the limits of our network by considering a large $M = 1000$ and a training set of $10^5$ images, using minibatches of $10^4$ examples (validation is performed on $10^4$ images). Even with all these simplification, the validation error still barely reaches $40\%$ before overfitting, as shown in the bottom left of Fig. 3. Despite the poor performance (as expected of this problem), we can still observe (bottom right of Fig. 3) that the parameter evolution of $a$ and $w^\|$ in very early steps of training is qualitatively in line with our theory. An interesting new feature in this case is the different duration of the regime for the two classes, where one class leaves the regime after only $\sim 20$ steps, while the other reaches a similar distribution around step 70. We suspect this behaviour to be due to the different "concentration" of the two classes, where one is rather homogeneous (mostly animals) while the other contains any other possible object. Remarkably, this asymmetry was not visible for the binary class case (not shown), where the evolution was closer to our theory, despite the network failing completely to generalize. We imagine that a different shape of data (e.g. two gaussians with different standard deviation) could account for this kind of effect in our simple model. However, this seems to be at the edge of the applicable cases for our theory, and a more complete treatment is needed to properly deal with the evolution in such complex scenarios.

## 3 Other material

**Code.** The code to reproduce all numerical results and graphs reported in this article can be found at https://github.com/phiandark/DynHingeLoss/. It consists of a single Jupyter notebook, based on Python 3 and requiring libraries numpy, scipy, tensorflow (1.xx), and matplotlib. All examples can be run in a few minutes on a moderately powerful machine. For more details, please see comments in the code.

**Time evolution.** An animation showing the training and validation error and parameters evolution for the same cases reported in Fig. 2 of the main text can be found on the same page. As discussed in Sec. 3.2 of the main text, the different behavior of the parameters is apparent, despite the similar final error. Moreover, the effects of overfitting can be noticed in the final phases of training.

## Footnotes

[1]These two features lead to mean-field interactions in which one parameter interacts weakly with all the others. In physical systems a particle instead interacts only with a finite number of other particles, hence the density field remains highly fluctuating. Only performing coarse-graining in space and time one can get hydrodynamic equations, see [2] for a rigorous presentation and [3] for a more general one.