[Reviews · NeurIPS 2020]

Review 1

Summary and Contributions: The authors found a case in which gradient descent dynamics in a two-layer neural networks can be described analytically in a very explicit form, in the mean-field limit (i.e. wide hidden layer) with infinitely many examples.

Strengths: The investigation of the transition from rich to lazy regime via the introduction of the parameter \alpha is very interesting. The paper makes assumptions to arrive to an explicit solution, but in the numerical and discussion section assesses very nicely going beyond these assumptions and comparing to toy, but realistic, simulation. Thus providing a very nice combinations of nice mathematical results and its relevance beyond the made assumptions.

Weaknesses: The used dataset if rather simplistic. The authors justify the choice of the dataset by the analytic sociability, which is ok and I buy it. But can they also give a hint where would the solution fail for other models of data? A lot of existing results are known for this same dataset and other learning models, e.g. without hidden units or with just a few hidden units, e.g. Refs. [11,13]. The authors could put their work in the context of these existing results.

Correctness: As far as I could tell the paper is correct.

Clarity: The paper is written clearly.

Relation to Prior Work: The related work section could be improved. For instance ref. [28] is cited for the use of a similar dataset, but isn't this dataset just the teacher perception as considered in statistical physics literature in dozens of cases, including e.g. ref. [11]. Related discussion could be included.

Reproducibility: Yes

Additional Feedback: I would suggest the authors state explicitely already in the abstract that their theory applies to the case of infinite dataset, as this is an important limiting factor. It is discussed nicely in the paper, including the discussion of the limitations, but one needs to read rather far in the paper to even realize this is the setting. ------- i have read the author's feedback and it addresses my suggestions. I am confident this paper will make a nice contribution to NeurIPS2020.


Review 2

Summary and Contributions: The authors solve exactly the gradient dynamics of two layer binary classification networks with a linear hinge loss. The problem is treated in a mean field limit (with a parameter interpolating between lazy and rich learning as proposed in the recent literature) and reduced to an effective single node problem. The data is seprable with spherical symmetry and the solution analytical. Results are also tested numerically which is useful for getting an idea of finite size effects that kick in at long times.

Strengths: An analytical solution of a special model of (shallow) neural networks. This is a useful benchmark. Interpolation between lazy and rich learning is also considered. It is also interesting that finite size effects are tested numerically.

Weaknesses: The mean field limit is not controlled rigorously. Probably the authors should be more precise when an approximation is used. For example passing from equation (1) to (2) is formal and should be stated more explicitly.

Correctness: Correct, as far as I can tell.

Clarity: I found that it is clearly written. In 3.1 I didnt get some of the details about the Gaussian distributions of initial conditions of a(0) w//(0) wperp(0). Are these assumptions or not ? This section was generally speaking less clear.

Relation to Prior Work: Yes

Reproducibility: Yes

Additional Feedback:


Review 3

Summary and Contributions: This is an elegant paper in which some advanced statistical physics methods are used to study the dynamics of learning in not trivial NN architectures. Of particular interest is the hydrodynamic treatment of the second layer. The authors specialize on the case of a linearly separable data and linear hinge loss. This allows them to solve analytically the dynamics. Several phenomena such as slowing down of the learning dynamics, rich and lazy learning, and overfitting can be observed in this simple setting.

Strengths: Though the results are mainly limited to the mean field (MF) infinite data and infinite width regimes, the paper provides an original view to the problem, which can hopefully open the way to improve over statistical physics MF techniques. The paper also serves to connect different languages and communities.

Weaknesses: The infinite data regime is perhaps the most relevant limiting factor. However the statistical physics methodology should be of interest for the ML community. The corrections to the MF limit are interesting though not conclusive. ************** I'm satisfied with the author's response.

Correctness: yes

Clarity: yes

Relation to Prior Work: yes

Reproducibility: Yes

Additional Feedback:


Review 4

Summary and Contributions: In the manuscript, the authors study in detail the training dynamics of a simple type of neural network: a single hidden layer trained to perform a classification task. Using mean-field theory, the authors developed this theory by treating the data set as data on a linearly discriminable hypersphere. Using the theory, several phenomena, such as slowing down of training dynamics, the crossover between rich and lazy learning, and overfishing, are revealed. The authors also verify the theory using MNIST dataset.

Strengths: This paper is excellent in that it provides a theoretical analysis of learning dynamics using mean-field theory for the discrimination problem, and is able to explain such phenomena as slowing down of training dynamics, the crossover between rich and lazy learning, and overfishing.

Weaknesses: It is unclear whether the input data assumption of a linear discriminable spherical distribution holds for the practical data such as a natural image dataset. The authors verify the theory using MNIST dataset as practical data. But the MNIST dataset is known to have a low-dimensional submanifold structure, so it seems relatively easy to ride on this assumption. The authors need to verify theory using more practical data, such as ImageNet dataset, or add a discussion of the validity of this data assumption.

Correctness: As mentioned above, I am concerned that only MNIST is targeted as an practical data experiment.

Clarity: The paper is well written.

Relation to Prior Work: Theoretical studies on learning dynamics other than mean-field theory should be cited and compared.

Reproducibility: Yes

Additional Feedback: ==== UPDATE AFTER AUTHOR RESPONSE ===== Thank you for your very careful reply. I decide to raise my rate of the manuscript because, this is because the authors have added experiments on the applicability of the theory to real data, such as ImageNet dataset. Thank you very much for your time-consuming experiments.

[Author Response · NeurIPS 2020]

We thank the reviewers for their valuable and constructive feedback. We are pleased that they generally appreciated our theoretical analysis in a simplified setting of phenomena such as slowing down of dynamics, rich and lazy learning, and over-fitting. We will implement all their suggestions which we think will greatly improve the quality of the text. We address specific points below.

**Reviewer 1** The discussion about related contributions was focused on recent works addressing the mean-field limit of infinitely wide one-hidden layer networks, since this was the theoretical framework we address. We agree with the **R1** that previous works mainly from statistical physics on few hidden units and teacher-student settings provide an interesting and relevant context for comparison. We will add a discussion. Concerning the dataset, in a more complex case the average over the dataset is not independent of the parameters (Eq. 9 does not hold). From numerical experiments, and ongoing work, we find that this is related to specialization of nodes, whose dynamics is governed by subsets of the whole dataset. We will add a brief discussion in the conclusion, as well as references to previous cases where specialization was observed and analyzed. We will also specify in the abstract that the analytical treatment holds for the infinite dataset case, and that corrections are studied heuristically and numerically.

**Reviewer 2:** The derivation of our results is based on theoretical physics methods, which are within reach of probability theory, with extensions of methods used in [17-22] and developed to rigorously treat the hydrodynamic limit in physics, e.g. *C. Kipnis and C. Landim, Scaling limits of interacting particle systems, Springer (2013); S. Serfaty, Coulomb gases and Ginzburg-Landau vortices, Zurich Lect. in Adv. Math., EMS (2015).* This makes us confident that all our results can be made rigorous, but we agree with **R2** that the non-rigorous steps should be stated and discussed more explicitly. We will do that when passing from eq. (1) to (2) [for which we assume the convergence of the empirical distribution to its average], and in the derivation of eq. (5) [for which we assume the convergence to the hydrodynamic limit of the dynamical process]. As for sec. 3.1, we will revise it to make it more clear. The parameters are normally distributed at initialization (see lines 71, 72), this implies that $a(0)$, $w^{\parallel}(0)$, and each component of $\mathbf{w}_{\perp}(0)$ in the space orthogonal to $\mathbf{w}^*$ are Gaussian distributed. We could have chosen different distributions, but we decided to focus on the Gaussian case as this is often used in practice.

**Reviewer 3:** We agree with the **R3** that considering the finiteness of the dataset is important, but it is also very challenging analytically. Our numerical experiments and heuristic arguments allow us to qualitatively understand the main new effects introduced by a finite dataset. An analytical treatment can be done in the case of a very large but finite dataset, in which one studies the Gaussian fluctuations of empirical averages around their means (these are small and of the order of (number of data)$^{-1/2}$), see eq. 18. A more complete treatment in which fluctuations of the empirical average are of the same order of the mean would be more conclusive indeed. However, this would require to control the entire distribution of the empirical averages, a quite difficult task in our case. One possibility to achieve this goal might be considering the simultaneous scaling limit $M \to \infty$ and $N \to \infty$ with a fixed ratio $M/N$, as done for example in Random Features models. This could provide a more conclusive picture but it is beyond the scope of the current work.

**Reviewer 4:** We chose a simple separable dataset to keep the model analytically tractable. We agree with the **R4** that realistic dataset are certainly more complex. However, we expect the dynamics of our simple model to share qualitative aspects with more realistic tasks, as in the early training a dataset can be roughly approximated as "two clusters with separate averages" and only later more complex features are learned. Several works have hinted at such progressive learning hierarchy *(e.g. A.M. Saxe, et al., Learning hierarchical categories in deep neural networks, (2013))*. Our numerical experiment was intended to illustrate our results for a realistic—yet still simple—dataset, since being limited to binary classification through a single hidden layer, one cannot reach good performance on real challenging tasks. Following the suggestion of **R4**, we report below (and will add to the SM) the results of experiments on CIFAR10 and ImageNet (analog to the ones of Fig. 4 on MNIST). The initial dynamics, while only poorly learning the tasks, does resemble the one we analyze in our simple model and found for MNIST (compare with Fig. 2 and Fig. 4). For ImageNet there is a difference in speed for the two classes, a feature that we could easily capture in our model by considering a different data distribution. Finally, we will add a broader discussion on theoretical studies on learning dynamics, in particular the ones on implicit regularization in which the dynamics of simplified models has also been analyzed.



[Meta-Review · NeurIPS 2020]

There were initially three positive reviews with high levels of confidence, and one negative review with low confidence, the latter of which has been revised upward after the author response. On the basis of the review reports, as well as my own reading of this paper, I would also evaluate this paper as positive, containing original and interesting contributions, including: - finding a case in which training dynamics of a single hidden layer neural network is analytically solvable in the mean-field limit, - exploring the lazy and rich regimes via controlling the parameter \alpha, - providing an O(1/M) correction to the analytical solution, leading to a spread in the dynamics, - providing an O(1/N) correction to the analytical solution, leading to overfitting. These items, as well as the insights gained therefrom, would compensate for the main weakness that the linearly separable model with spherically-distributed samples is simplistic (which is for the sake of analytical tractability). I would thus recommend acceptance of this paper. Minor points: "asses" should read assess. \theta(.) appearing in equations (6) and (7) is undefined.